# The Relevance of Mass Spectrometry Analysis for Personalized Medicine through Its Successful Application in Cancer “Omics”

**DOI:** 10.3390/ijms20102576

**Published:** 2019-05-25

**Authors:** Cristina Alexandra Ciocan-Cartita, Ancuța Jurj, Mihail Buse, Diana Gulei, Cornelia Braicu, Lajos Raduly, Roxana Cojocneanu, Lavinia Lorena Pruteanu, Cristina Adela Iuga, Ovidiu Coza, Ioana Berindan-Neagoe

**Affiliations:** 1MEDFUTURE -Research Center for Advanced Medicine, “Iuliu Hațieganu” University of Medicine and Pharmacy, 4-6 Louis Pasteur Street, 400349 Cluj-Napoca, Romania; crisciocan@gmail.com (C.A.C.-C.); da.buse@gmail.com (M.B.); diana.c.gulei@gmail.com (D.G.); pruteanulavinia@gmail.com (L.L.P.); cristina.iuga@gmail.com (C.A.I.); 2Research Center for Functional Genomics, Biomedicine and Translational Medicine, “Iuliu Hațieganu” University of Medicine and Pharmacy; 23 Marinescu Street, 400337 Cluj-Napoca, Romania; ancajurj15@gmail.com (A.J.); braicucornelia@yahoo.com (C.B.); raduly.lajos78@gmail.com (L.R.); cojocneanur@gmail.com (R.C.); 3Department of Pharmaceutical Analysis, Faculty of Pharmacy, “Iuliu Hațieganu” University of Medicine and Pharmacy, 6 Louis Pasteur Street, 400349 Cluj-Napoca, Romania; 4Department of Oncology, “Iuliu Hațieganu” University of Medicine and Pharmacy, 34-36 Republicii Street, 400015 Cluj-Napoca, Romania; 5Department of Radiotherapy with High Energies and Brachytherapy, Oncology Institute “Prof. Dr. Ion Chiricuta”, 34-36 Republicii Street, 400015 Cluj-Napoca, Romania; 6Department of Functional Genomics and Experimental Pathology, Ion Chiricuțǎ Oncology Institute, 34-36 Republicii Street, 400015 Cluj-Napoca, Romania

**Keywords:** omics technology, mass spectrometry, personalized medicine, biomarkers

## Abstract

Mass spectrometry (MS) is an essential analytical technology on which the emerging omics domains; such as genomics; transcriptomics; proteomics and metabolomics; are based. This quantifiable technique allows for the identification of thousands of proteins from cell culture; bodily fluids or tissue using either global or targeted strategies; or detection of biologically active metabolites in ultra amounts. The routine performance of MS technology in the oncological field provides a better understanding of human diseases in terms of pathophysiology; prevention; diagnosis and treatment; as well as development of new biomarkers; drugs targets and therapies. In this review; we argue that the recent; successful advances in MS technologies towards cancer omics studies provides a strong rationale for its implementation in biomedicine as a whole.

## 1. The Advent of “Omics”: Transitioning Mass Spectrometry from Cancer to Medicine

One of the most important aspects of medical research is the establishment and development of better therapeutic options for patients. Because of the individual variation of the human proteome; precision and personalized medical strategies are of relevance; based on the specific protein or molecular signatures and cellular context [1]. Due to the various applications of mass spectrometry (MS); this analytical technique has been newly considered the standardized foundation of oncological approaches identifying therapeutically-significant players in clinical laboratories (as proteins profiling; endogenous or exogenous molecules and their metabolites; etc.); the first essential step for building a successful individual treatment.

The “Omics” represent a set of complex technologies used to study the roles; relations; and activities of the different types of molecules that compose the cells of an organism. The aim for any biological sample is the detection of patterns or specific genes (genomics); mRNAs (transcriptomics); proteins (proteomics) and metabolites (metabolomics) (Figure 1). The genome contains the basic information that can be inherited; or it can be modified throughout a person’s lifetime. Therefore; it contains both coding and non-coding DNA regions that are transcribed into various types of RNA species. All RNA types constitute the transcriptome; this entails that the transcriptome also contains both coding and non-coding RNA species. The non-coding RNAs fulfill various roles in the cell; from ribosomal constituent or transportation to gene expression regulation. The coding RNAs are translated into proteins. All the proteins within a living cell are grouped under the name of proteome. The functional unit of a cell is its proteins; determined by its structure which does not remain constant. Proteins are unique in that their structure can be post-translationally modified; thereby altering their function. This means that proteins have more than one state which through inactivity (indirectly) or activity (directly) determine their functional use within the cell. The most common post-translational modification (PTM) that occurs is phosphorylation; turning certain enzymes into an active or functional for cellular energy transfer purposes. Regardless of the post-translational modification; the role that non-functional proteins have has not been completely elucidated. The various proteins and their isoforms have highly complex interactions with each other. Through the enzymatic interaction of various types of proteins; metabolic pathways are built. These pathways determine how endogenous or exogenous molecules will be processed in the living cell. The molecules resulting after the enzymatic reactions between proteins are named metabolites. Thus; omics technologies can be used in the medical field to understand the etiology of diseases. In turn; this emphasizes their potentially vital role in screening; diagnosis and prognosis; with some applications in drug discovery; evaluation of their toxicity and efficacy.

Genomics relies on the studies of the structure; function and relative expression of a gene in an organism [2,3]. Total DNA quantifies as the genome of a cell or organism. The genome is generally divided in non-coding genes; meaning not translatable into proteins; which accounts for approximately 98% of the genome [4] and 2% of coding genes [5,6]. It is known that the human genome has 3.2 billion bases [7] and about 30,000–40,000 protein-coding genes [8,9,10].

Functional genomic analysis comprises both genomics and transcriptomics; using DNA-seq and RNA-seq to describe genes and transcripts with relevance to protein prediction and their functionalization or interconnection to different patients [11,12]. Transcriptomics refers to the study of RNA in a cell or organism. Microarray technology utilizes the packaged mRNA as the transcriber of genes which are then measured and quantified through their relative expression. In essence; microarray represents a summary of gene activity through the changes in mRNA abundance [13,14,15]. Next Generation Sequencing (NGS) technique is used to identify genetic variants; genotypes; mutations; insertions; deletions; structural and genomic variants; single nucleotide polymorphisms (SNP) [16]. SNPs occur in the DNA sequence and represent the common variations relying on the substitution of a nucleotide with another one. SNPs present a central function in diseases with a genetic background and also in pharmacogenomics where investigation targets the individual patient responses to drugs [17].

High-throughput RNA sequencing (RNASeq) has appeared as an attractive and cost-effective technique for transcriptome profiling [18,19]. RNASeq is used to study gene expression and identify new RNA species. This technique offers less background noise and a higher dynamic range for detection. Also; it facilitates the discovery of the sequence identity; important for unknown genes analysis and new transcript isoforms [20]. Another technique used is comparative genomic hybridization (CGH) which analyzes chromosomal deletions and duplications present in whole genome.

Proteomics refers to all proteins present in a specific cell; a pattern that is highly dynamic between cell types and environment response. The proteome is controlled by a wide range of factors; such as the complexity of protein synthesis; degradation and modification; which effects protein localization; function and connection between partners. Proteomics links our understandings of complex biological systems through the expressible; identifiable; modifiable intermediary because proteins represent quantifiable specific characteristics of disease [21]. Moreover; protein-protein interactions and protein post-translational modifications demonstrate the complexity at which protein’s function [22]. These post-translational modifications act like cellular control or regulation mechanisms affecting the properties therein the functions of the proteins and are exhibited in cancer development [23]. To ensure the detection of low abundance proteins; provide quantitative and qualitative analysis of proteins; separation of complex protein mixtures is required. To increase the possibility to visualize low abundant proteins and maximize the coverage of the proteome; it is useful to prepare standardized partial proteomes encompassing complexity of protein properties differentiation [24]. An effective detection methodology for increasing the coverage of the proteome is fractionation; which provides molecular information about the state of a tissue and the distribution of its protein components [25]. Subcellular fractionation represents an approach for all cell types and tissues; enriching the analysis of intracellular organelles and low abundant multi-proteomic complexes [21].

Proteomic analysis is considered a useful approach to identify the targets of bioactive compounds; drugs; chemotherapy [26,27] and to explain their molecular mechanism of action [28]. It provides insight into the mechanism of different diseases and an understanding of their pathogenesis; which is imperative for establishing standardized diagnosis [29]. Given all of this; proteomics has the largest applicability in the cancer field; presenting a potential in discovery of new biomarkers that may enhance the diagnosis of different types of cancer [30]. Through its application towards cancer research; it will demonstrate an analytical approach that could be applied to all diseases. Proteins symbolize the most tangible and attainable product of all the omics at this moment when compared to genomics or transcriptomics which are abstractly important for their molecular mechanisms of regulation or modulation.

There are two major strategies applied to proteomics: top-down based on intact protein level analysis or bottom-up based on peptide level analysis. Bottom-up proteomics is distinguished by proteolytic digestion before mass spectrometric analysis; which allows for identification of proteins and determining amino acid sequences or post-translational modifications. This strategy combines MS with techniques like gel electrophoresis for protein purification or liquid chromatography for proteolytic peptides or protein fragments separation [31]. It comes as no surprise that a frequently used method representative of this bottom-up strategy is liquid chromatography-high resolution tandem mass spectrometry (HRMS). When bottom-up analysis is performed on a mixture of proteins it is called shotgun proteomics [31]. Conversely; the top-down proteomic strategy does not use proteolytic digestion; but intact proteins; fractionated by molecular weight or immunoprecipitation. The resulting fractions are analyzed by high resolution accurate mass MS/MS instruments with online chromatography. The intact proteins are ionized by electrospray ionization (ESI) for mass measurement; the proteins could be “trapped” in Fourier transform ion cyclotron resonance (FT-ICR) or Orbitrap mass analyzers. A frequently used method representative of the top-down strategy is protein ionization by matrix-assisted laser desorption ionization (MALDI) and mass measurement by time-of-flight (TOF) mass analyzer [32]. It should be noted that there have been more recent approaches developed to investigate proteins that combine the aforementioned strategies. One such example is middle-down proteomic strategy featuring size-dependent protein fractionation technique and robust but restricted proteolysis method for interrogating high-mass proteomes. Wu et al. use protease OmpT to perform this robust but restricted proteolysis of an entire complex proteome [33].

Post-translational modifications can be widely defined as covalent processing events that change the properties of a protein by proteolytic cleavage or by addition of a modifying group to one or more aminoacids [34]. Thus; there are many different types of post-translation modifications that can occur; but the most common and studied ones include: phosporylation; acetylation; methylation; glycosylation; Glycosylphosphatidylinositol anchoring; sulfanation; disulfide bond formation; deamidation; ubiquitination or tyrosine nitration. Of all of these; phosphorylation is the most frequent [35] and not coincidentally; the most studied. It should be noted that phosphorylation not inherently linked to protein functionality because protein kinases and phosphatases work independently; and only by their dynamic balance do they regulate function.

Metabolomics is the study of global metabolites profile in the cell; tissue or organisms; setting specific conditions for research experiments [36,37,38]. Metabolomics refers to the activity of the small molecules (<10 kDa) that is achieved by their active state in living cells through their life cycles. Meanwhile; it offers information regarding the physiological status of the cells and allows researchers to understand how a metabolic profile of a complex biological system is modified in response to stress [39]. The metabolomics field includes the smallest domain (~5000 metabolites); along with different types of biological molecules; making it more complex from a physical and chemical point of view than the other ‘omics’ [17]. Metabolites (e.g., lipids; vitamins) are responsible for the transmission of energy in cells resulted by the interaction with other biological molecules [40]. Therefore; metabolome is formed from a wide range of different chemical structures that are inherently greatly variable and time dependent. One important consideration that should be emphasized specific to the metabolome is the fact that the metabolites are time-dependent meaning they do not stay in that state (or isoform) indefinitely.

It is important to achieve qualitative and quantitative information regarding metabolites that occur under normal conditions to be able to detect abnormalities due to changes in environmental factors [41,42]. The applicability of mass spectrometry in metabolomics field consists of the study of the effect of drugs; toxins and different diseases based on metabolite levels. This entails focusing on metabolic pathways and measuring fluxes [43,44]. So far; the investigations have offered information about different disease types (such as breast; prostate; colorectal; gastric cancer; kidney and cardiovascular diseases); effects of toxicology and nutrition; and metabolic fluxes [45].

The simplified representative relationship between omics and corresponding analysis are presented in Figure 2. Each of the omics are very broad fields; in this diagram we choose the most common or frequently studied analytical investigations.

Genomics; transcriptomics; proteomics and metabolomics are interdisciplinary research areas and together can accelerate studies for early diagnosis of many diseases such as cancer; diabetes [46]; cardiovascular; neurodegenerative (Alzheimer and Huntington disease); phenylketonuria [47]; asthma [48]; intestinal autoimmune disorder (celiac disease); developmental disorders [49]; inflammatory bowel disease and obesity. Moreover; this collaborative progression of omics fields improves drug development and most importantly; can lead to a made-to-measure treatment option for every patient summarized in Table 1; this indirectly leads to a better treatment outcome and quality of life for the patients [15,50,51].

In the future; omics technologies may offer us the possibility to improve approaches that intend to be predictive; preventive and personalized [57,58]. Specifically; the representative analytical tool used to highlight the omics technologies is mass spectrometry primarily utilized for proteomics and metabolomics analysis. Through mass spectrometric quantification; it is possible to identify and characterize the target proteins. This improves our understanding of the structure-function relationship of synthetic compounds for early-to-advanced drug metabolism and pharmacokinetics studies [59]. In addition; an important perspective is gained by investigating potential biomarkers because it provides information about the disease progression and drug action [60]. Mass spectrometry; besides being the en vogue technique; is a robust and efficient tool in the clinical research. This technique perfectly fits the requirements of research guidelines and can be easily translated into a routine or standardized methodology. For a proper validation of an analytical method; it is necessary to demonstrate through precision and accuracy tests; the quantification of the compound of interest from the biological matrices.

In order to achieve this new vision for personalized medicine; an important role is attained by mass spectrometry which allows a rapid and early diagnosis; specific therapies for patients and real-time therapeutic drug monitoring [17].

The purpose of this paper is to present the successful applications of mass spectrometry in cancer research to emphasize the impact mass spectrometry-based techniques can have for the future of personalized care.

## 2. Expanding the Horizon for Mass Spectrometry

### 2.1. Basic Concepts Regarding MS

Mass spectrometry (MS) is a comprehensive system for identification and quantification of organic molecules; representing a sensitive; specific and selective method [61,62]. The sensitivity of current mass spectrometry systems is translating into the possibility of detection; identification and quantification; based on mass-to-charge (*m*/*z*) ratio; of analytes at concentrations in the attomolar range (10–18). This is an important feature of the technology; which ensures the ability to accurately measure an analyte in the presence of other interfering products (such as degraded products that may be present in the processed sample matrix). Despite the difficulty of studying proteins from biological systems (e.g., cells; tissue; organism) due to the dynamic nature of protein expression; this system obtains unique fingerprints correlated to proteins/peptides from known databases facilitating identification of unknown compounds [63]. The specific validation parameter of the analytical method should demonstrate the unequivocal assessment in the presence of other compounds: particularly if the analyte detection follows the International Conference on Harmonization requirements; if the substance is active; or if it is pharmaceutical formulation. Conversely; where ICH guidelines are not applicable; mass spectrometry determination can also be focused on the characterization of biological matrices. Consequently; the presence of non-specific and specific matrix-related interferences (as endogenous lipids; rheumatoid factor; heterophilic antibodies; endogenous protein analogs; degradation products; catabolites; etc.) should be minimized during the development of extraction procedure [64].

Mass spectrometers are being used in medicine for establishing the prognosis and stage of a disease [60]; discovery of next-generation cancer biomarker panels [57]; and assessment of the distribution or biochemical influence of chemotherapeutic agents [65]. Through this method; the protein contents of specific cellular compartments can be identified; thereby allowing the understanding of their function; their localization in the cells and their dynamic changes. Proteomics is used to distinguish the function of proteins through their physio-chemical properties; making it an effective way to study the complexity of mechanisms involved in the regulation of cellular functions [66]. MS allows for determining the organic molecules at very low concentration levels [67] with high sensitivity [61]. Many molecules can have the same mass-to-charge ratio despite having very different chemical structures. According to mass-to-charge ratio (*m*/*z*) of the molecular ion; isotope patterns and fragment peaks; the molecular mass and molecular formula of the molecules can be identified [67]. It provides a specific; sensitive response that is used to detect and integrate a peak in a simple one-dimensional chromatographic separation of the sample [68]. For cancer research; mass spectrometry imaging has three essential areas benefitting from its application: next-generation prognostic and therapeutic biomarkers due to the possibility of creating a chemical mapping morphological of-interest regions; evaluation of distribution and biochemical effect of chemotherapeutic agents offering an understanding in anti-cancer drug efficacy; and classification of tissue types based on molecular ion patterns [65]. This should emphasize that the use of MS has been limited to primarily cancer but does not mean it has to remain constrained there. The main steps of mass spectrometry-based proteomics are sample processing; data acquisition; and data processing; which can be seen Figure 3.

The proper preparation of biological samples is an important and critical step in the workflow; having the greatest impact on data reproducibility. Protocols differ depending on the biological matrix and type of experiment; which includes the aim of the investigation. There are major advantages to having several variations of mass spectrometry because firstly; implementing multiple pre-treatment protocols of the biological samples improves our understanding and experimental efficacy by reducing the protein abundance the analyte; secondly; the equipment permits a direct scanning and analysis of the sample; and lastly; different variations allow for cross-validation of the data. For mass spectrometry data; public processing repositories/libraries such as PeptideAtlas (www.peptideatlas.org) and SRMAtlas (www.srmatlas.org) could be consulted or generated by specialized software.

The special advantage offered by this versatile technique is exemplified by its wide range of applicability: data regarding the structure of interest molecules; accurate determination of molecular mass of peptides; proteins; and oligonucleotides; protein expression level profiling; amino acid determination in different types of samples (polypeptides and proteins); quantification of specific fragments separated by chromatography and capillary electrophoresis; or quantitative analysis of metabolites from different types of biological samples (blood; urine; and saliva). Due to its vast applicability; robustness and reproducibility as well as current transparent policy of laboratory work procedures; MS use in the medical field is imperative; both research and analysis routine.

The most common mass spectrometry configurations or technologies are presented in Table 2.

### 2.2. Small Molecule Identification; Quantification and Monitoring

For a proper highly selective method to be applied on liquid chromatography/mass spectrometry systems is necessary to accurately optimize the detection selectivity parameters (options related to the MS equipment); concurrently with separation selectivity components (mobile phase; stationary phase; temperature; flow; etc.). It is the combination of these two techniques; liquid chromatography for separation of compounds based on physico-chemical properties and mass spectrometry for differentiation based on mass-to-charge ratio; that generates an extremely powerful analytical tool [69].

One application for LC-MS/MS is the quantification of small molecules from biofluids. For sample processing; preparation and separation of biological matrix components from the analyte of interest; it is necessary to perform liquid-liquid extraction and solid phase extraction to achieve high performance with LC-MS/MS and filtration of particles [70]. A pre-concentration step is mandatory to provide suitable sensitivity for MS detection. The ion source; the mass analyzer and detector are important components of MS. Injection of the samples into the mass spectrometer relies on conversion of the molecules into either cations or anions in the ion source. In the mass analyzer; separation occurs according to their mass/charge (*m*/*z*) ratio in a vacuum under the influence of electric or magnetic fields. The final step is detection. For analysis; LC uses reversed phase chromatography which successfully separates a wide range of samples with higher sensitivity; reduced run times; shorter elution times and lower solvent consumption. For detection; triple quadrupoles are used for quantitative analysis due to their high sensitivity and selectivity [31,70,71,72].

The use of chemotherapeutic drugs has become the standard; efficient treatment option due to the higher incidence of cancer. However; these cytotoxic drugs used to treat different cancer types; also affect normal cells having a toxic; carcinogenic; mutagenic and teratogenic potential even at low concentrations [73,74]. All of these agents once administered interact with various proteins or metabolites which present a ripe source for biological quantification.

There are also therapeutic chemotherapy strategies based on combinations between different drugs for each cancer type [75]: for non-small cell lung cancer; combinations include cisplatin-associated compounds; for colon cancer 5-fluorouracil; leucovorin and irinotecan are used in combination; for breast cancer; an anthracyclines; carboplatin and paclitaxel combination is used; and lastly; in oral cancer a combination panel of methotrexate; cisplatin; carboplatin; 5-fluorouracil; paclitaxel and/or docetaxel is used [76]. The major side effects of these drugs are associated with toxicity that can affect normal bone marrow stem cells and healthy epithelial cells [77]. This is important for mass spectrometry because it demonstrates that these agents interfere with both normal and tumor cells; providing an opportunity for analysis and characterization. Case in point; Allardyce et al. utilized electrospray ionisation mass spectrometry and tandem mass spectrometry for data analysis to determine the interaction of cisplatin with transferrin [78]; more specifically; identifying the cisplatin binding site; the amino acid residue involved in metal binding; on the protein transferring.

Cyclophosphamide; an oxazaphosphorine derivate; is another widely used cytotoxic drug in various neoplastic diseases; autoimmune disorders; chronic lymphocytic leukemia and lymphomas [79]. Cyclophosphamide is an inactive drug. Through enzymatic activation; phosphoramide mustard and acrolein are formed; which are the active alkylating agents with toxic and therapeutic effects. In addition; methotrexate (MTX) is another known chemotherapeutic drug for a wide range a neoplastic diseases such as acute lymphoblastic leukemia [80]; non-Hodking lymphoma; cerebral tumors [81]; bronchial; breast [80]; ovarian [82]; bladder cancer [83]; osteogenic sarcoma [84] or as an immunosuppressive agent to treat inflammatory bowel disease [85] and rheumatoid arthritis [80]. This folate antagonist is a mutagenic and teratogenic anticancer drug which inhibits cellular function; prevents cell replication; leads to DNA damage; blocks enzyme activity and causes cell death due to binding of the active site of dihydrofolate reductase [86]. An essential method to determine and monitor the levels of the chemotherapeutic agents cyclophosphamide and methotrexate in human serum is LC–MS/MS using a C18 analytical column [87,88]; which requires the use of “clean-up” procedures. The most common “clean-up” methods are: solid-phase extraction using specific cartridges and protein precipitation using trichloroacetic acid as solvent [89,90]. Using “clean up” procedures can greatly impact the recovery range and enhance sensitivity of detection; for example; solid-phase extraction for LC-MS/MS coupled with ESI lead to a recovery rate range of 99.38 to 103.21% and a detection limit is 3 ng/mL [90].

Additionally; the use of organic solvents for the extraction of proteins and peptides from serum is efficient and facilitates the identification and comparison using different mass spectrometry approaches [91]. Chertov et al. developed such a method; under denaturing conditions using acetonitrile containing 0.1% trifluoroacetic acid; for the extraction of peptides and low molecular weight proteins from serum samples. Using an older approach; called Surface-enhanced laser desorption/ionization-time of flight mass spectrometry (SELDI-TOF MS); they were able to detect two significantly reduced markers in mice with B cell lymphoma tumors; Apolipoprotein A-II was identified as one of these markers [91]. It should be noted that liquid extraction with the use of organic solvents is used more frequently in Metabolotics. More specifically; if the metabolites are moderately polar; non-polar or hydrophobic; organic solvents must be included. It offers a unique advantage when compared to water in that more diverse metabolites can be extracted. For example; when implementing a two-phase solvent system with water-methonal-chloroform both polar; hydrophilic or non-polar metabolites can be extracted simultaneous. Furthermore; organic solvent-based extractions also offer the advantages of: easy solvent evaporation; absence of precipitate salts and increased stability of extracted metabolites [92]. Last but certainly not least; the organic solvents used for extraction are compatible with GS-MS; LC-MS; HPLC and capillary electrophoresis [92].

Mass spectrometry is a simple; rapid and sensitive method used to monitor the concentration of immunosuppressive drugs in biological samples offering higher sensitivity and selectivity. Improvement of the therapeutic index can be obtained by combining the specificity of an antibody toward a certain cell surface antigen with the toxicity of a small molecule drug; such as tubulin inhibitors and DNA cross-linking agents. The aim is to develop a specific antibody-drug conjugate. This relies on the identification of a cell surface protein that is selectively expressed in tumors and coupling it with a highly specific monoclonal antibody (mAb) appropriate to the linker–toxin combination [93].

Sirolimus is one such immunosuppressive drug which is used to prevent transplant rejection and to treat immune mediated diseases [94]. Sirolimus inhibits mammalian target of rapamycin (mTOR); a protein kinase and member of the phosphatidylinositol 3-kinase family which are involved in survival; cell growth and proliferation [95]. Through its mechanism of action; Sirolimus bind to the cytosolic protein FKBP12 and inhibits mTOR by reducing DNA and protein synthesis [96]. Sirolimus blocks interleukin-2 and inhibit the proliferation of T-cell [97] by preventing progression of the cell cycle from the G1 to the S phase [98]. FKBP12 protein is involved in modulating inflammatory response and regulating cellular processes associated with growth [99]; differentiation and angiogenesis [100]. Furthermore; it is responsible for inhibition of T cell and B cell proliferation [101]. This culminates in the necessity to identify and quantify Sirolimus in patient samples to better monitor its effects. On that note; two interesting recent articles used MS-based approaches for therapeutic monitoring; which could be applied to the boarder medical field. Firstly; to quantify the amount of Sirolimus from whole blood LC-tandem mass spectrometry using solid-phase extraction was performed [102]. Secondly; HPLC/electrospray-mass spectrometry was used to quantify sirolimus and its metabolites from the blood of kidney graft patients [103].

In terms of targeting novel cancer proteins using antibodies; Surface Antigen In Leukemia (SAIL) is exemplary for its proteomic identification using LC-MS/MS. SAIL represents a target for hematologic malignancies being overexpressed in tumors samples compared with normal ones. In patients with chronic leukemia lymphoma (CLL); SAIL is 90% overexpressed compared with 29% of acute myeloid leukemia (AML) and 3% of multiple myeloma (MM) where low abundance of both malignant cells suggests an underestimation of protein expression levels [93]. Using bottom-up proteomic strategy; SAIL expression was not detected in other types of diseases; like colorectal; ovarian or lung cancer and their respective normal adjacent tissues [93]. Through quantitative protein analysis; biomarkers that differentiate a drug resistant tumor cell from to a drug sensitive tumor cell can be identified. Biomarkers are early predictors of cancer development; including MM; and are easy to identify from plasma or serum because they contain large quantities of proteins. In addition to biomarkers; immune response is involved in detection of cancer because it is believed to be auto-antibodies against cancer cells are reproduced in large quantities for detection.

### 2.3. Variations of MS and Their Associated Applications

Matrix-assisted laser desorption/ionization time-of-flight mass spectrometry (MALDI-TOF) is a top-down proteomics technique that was coined in 1980s for peptide and protein analysis by Franz Hillenkamp and Kochi Tanaka. In addition; this technique is also used to characterize different tissue types by direct exposure [104]. For this method; a relatively small amount of sample is used; 1 μL; that contains the peptides and proteins [105]. The dried samples are placed on target plate and then vaporized to release ion protein molecules which are speed up in an electric field. According to their mass/charge ratios (*m*/*z*); ions with lower ratio are accelerated to higher rate and arrive before ions with a high ratio in detector. Separations occur in a vacuumed electric field and the ions are generated by a single laser pulse; thereby; providing short analysis time per specimen. MALDI-TOF MS resolution can be improved by using an electronic mirror or delayed extraction. Despite the electronic mirror enhancing resolution; it still has a drawback because it gradually reduces transmission of ions to the detector; meaning that some of the analysis continues to be represented in a linear ion mode. The purpose of delayed extraction is to introduce a delay after sample vaporization to apply electric potential. Smaller molecules require shorter delay times; compared to larger molecules. The advantages of delayed extraction are increased resolution and this set-up of a delayed time optimizes the analysis for a specific *m*/*z* range [106]. The microchannel plate is the standard detector that operates as an electron multiplier for arriving ions. The response of the detector relies on the number of ions that get to the detector and ion rate. A stronger detector response is produced by the fast ions with low *m*/*z* ratio compared to slow ions. In conclusion; MALDI-TOF MS is one of the few technologies that guarantees a sensitive detection for small molecules [106].

MALDI—MS presents a wide range of new applications. Coupling MALDI with linear ion trap Orbitrap or quadrupole-time-of-flight (QTOF) facilitates generating lower mass range (1000 Daltons) [107]. MALDI-MS has been applied in various therapy-driven areas; such as lipidomics; peptides; drugs and metabolites from biological samples [108]. Aforementioned; this technology is used to differentiate tissue types to classify tumor grade as well as tumor origin [109] and identify metabolism pathways. For example; Pirman et al. used MALDI coupled to a QTOF mass spectrometer to investigate two non-small-cell lung cancer (NSCLC) cell line treated with eicosapentaenoic acid (EPA). Tissue lipid metabolism of eicosapentaenoic acid from Kras transgenic mouse lung tumors were analyzed and characterized by adapting techniques relying on LC-MS/MS and MALDI-MS were used [107].

Wang et al. determine in their study the serum peptides biomarkers for colorectal cancer by MALDI-TOF MS combined with Weak cation exchange magnetic beads (MB-WCX). MB-WCX is a method that can capture low abundance proteins or peptides in serum samples. The identification of these peptides was performed using LC-MS/MS. Notably; MALDI-TOF MS and LC-MS/MS were applied to analyze; diagnose human disease and identify potential biomarkers of health status using biosamples such as serum/plasma; saliva and urine [110].

In their study; Kan et al. present a mouse model of persistent HBV infection. They used RNAseq and 2D–MALDI-TOF/TOF to investigate HBV-specific proteomic and transcriptomic signatures. Thus; they analyzed in LX2 human hepatic stellate cell line several differentially expressed proteins. Data analysis provided information on proteins that are involved in oxidative stress responsible for the liver fibrosis development. Furthermore; RIG-I-like receptor signaling pathway was shown to be a central pathway that undergoes changes during HBV-related fibrosis. The successful aim was to identify important gene transcription and expression profiles for therapeutic targets used for diagnosis and understanding the underlying liver fibrosis molecular mechanism. More specifically; CAT; PRDX1; GSTP1; NXN and BLVRB were demonstrated to be associated with oxidative stress among the differentially expressed proteins in the RIG-I-like receptor signaling pathway; moreover; results were validated by Western blot and RT-qPCR [111].

Qin et al. tried to uncover non-invasive biomarkers for early detection of gastric cancer based on glycomic analysis. Combining ethyl esterification derivatization of glycans with MALDI-TOF MS; provides a vast serum glycomic analysis able detect neutral N-glycans and ethyl esterified sialic acids. According to their characteristic structures; several N-glycans were identified as biomarkers with the ability of distinguishing the early stage of gastric cancer from healthy patients and monitoring the progression of gastric cancer [112].

Swiatly et al. used MALDI-TOF-MS in their study to characterize serum from patients with ovarian cancer compared to healthy control group; thereby; identifying potential biomarkers. Thus; only through this technique; they identified four potential ovarian cancer serum biomarkers: complement C3; kininogen-1; inter-alpha-trypsin inhibitor heavy chain H4 and transthyretin [113].

In another experiment; Wang et al. used MB-WCX coupled with MALDI-TOF MS to compare the serum peptidome profiles of advanced lung adenocarcinoma patients treated using first-line platinum-based pemetrexed chemotherapy [114]. Using the MS-based technique and ClinProTools software; these profiles were used to determine potential predictive peptide biomarkers and develop a predictive peptide model for accurate group discrimination. Validation samples were then classified into “good” or “poor” outcome groups based on the clinical outcomes of objective response rate; disease control rate; progression-free survival and overall survival. In this validation set; only OS was not statistically strong enough for prediction; the other clinical outcomes were statistically strong predictors of being a part of the “good” group vs. the “poor” group. Thus; the predictive peptide model based on four distinct *m*/*z* characteristics was determined by the clinical outcomes of patients on first line pemetrexed plus platinum treatment. Lastly; eight potential peptide biomarkers were identified based on eight mass [Da] peaks significantly differentially expressed in the patient training set; of these; only two were the peptide sequences and proteins were known; uridylate kinase and Glucosamine--fructose-6-phosphate aminotransferase [114].

Akpinar et al. evaluate the differentially expressed proteins between parathyroid adenomas (PA) and parathyroid hyperplasia (PH). In terms of methodology; they combined 2D gel electrophoresis (2DE) with MALDI TOF-TOF MS. According to their expression levels; they identified a total of 40 proteins whose levels have changed: fourteen overexpressed in PH and 26 overexpressed in PA. The results particularly suggest that proteins overexpressed in PH were mitochondrial. Thus; this emphasizes the demand for more research into the role mitochondrial activity plays in disease and perhaps the discovery of novel mitochondrial marker used for differentiating PH from PA [115].

Mass spectrometry imaging (MSI); the newest technique; is used to investigate the molecular distribution of metabolites; proteins and lipids without labeling. This novel ionization technique allows to analyze the tissue by desorption electrospray ionization (DESI); where the corresponding steps in sample preparation are reduced. Most interestingly; this technique has a great applicability in intraoperative tumor assessment. Meanwhile; it facilitates characterizing the differences between cancer and non-cancer tissue based on lipidomic information and the correlation of lipid distribution with tissue morphology. Distinctive lipid profiles that are associated with different forms of cancer have been investigated with this technique and the results correlate with histopathological examination [116]. Inevitably; the aforementioned technologies were combined into MALDI-MSI; which was introduced in 1994 [117]. The aim of this technology is to localize different molecules allowing for evaluation of their spatial and temporal distribution in tissue; organs or whole body [118]. What’s more; MALDI-MSI can be used to localize other molecules such as lipids and drug components. In the past years; this technique has been developed for drug distribution [119] with application in therapeutic agent imaging of low-molecular-weight compounds. To enhance the localization of drug metabolites and detection of pharmacologically active compounds; high resolution MSI is implemented. All of this is exemplified by Vegvari et al. introducing thier MALDI-MS imaging approach to investigate the localization of tamoxifen in different ER-positive/negative breast cancer tumor sections. The results show that the distribution of tamoxifen in ER-positive/negative tumor cells is reduced in comparison with stroma in ER-negative samples [120].

To best characterize the variations within the same protein structure and; in essence; to investigate post-translational modifications; the protein ideally needs to remain intact meaning not digested or fragmented. This requires top-down proteomic strategies for analysis. Using electrospray ionization (ESI) and quadrupole (Q) mass analyzer is one such alternative for serum biomarker identification using liquid chromatography for sample fractionation and ESI-Q technology [121]. To analyze the target samples with high performance liquid chromatography (HPLC); firstly, the solution is nebulized under atmospheric pressure and then exposed to a high electrical field that generates on the droplet’s surface a charge. Droplets become much smaller due to the evaporation before getting into the analyzer. Through HPLC-ESI-QMS; peptide fragmentation can be performed from peptide sequences. Utilizing the existing data analyses; algorithms and databases ensures peptide characterization at the aminoacidic level [122]. Despite not falling under the umbrella of proteomics; Weiskopf et al. exemplified the powerful characterization capacity and bio-complexity threshold of QESI-MS. QESI-MS was explicitly used to determine oligosaccharide composition; sequence; branching and linkage; providing a unique proof of concept. Furthermore; to demonstrate the bio-complexity that ESI-MS can characterize in given samples; complex oligosaccharides from hen ovalbumin were combined with high mannose oligosaccharides from RNase B (of compositions GlcNAc2Man5–GlcNAc2Man9) [123].

Surface-enhanced laser desorption/ionization mass spectrometry (SELDI-MS) is a top-down proteomics approach; introduced in 1993 and used for the quantification of small peptides and proteins relative to large proteins. SELDI-TOF MS is an older alternative to MALDI-TOF MS [124]. In this technique; target surface is specifically transformed to contain ion-exchange; normal-phase; hydrophobic or metal chelate functional groups. The surfaces can also be modified with antibodies; proteins with adequate binding properties or DNA structures. According to protein properties; surface area; binding capacity and wash buffers; a small proportion of proteins are selectively attached on the target surface [104]. These two methods; MALDI and SELDI; are different. For example; in MALDI the sample is mixed in solution with a matrix molecule and then deposited on a surface to dry. In this way; after solvent evaporated; sample and matrix molecules co-crystallize. Conversely; in SELDI; the sample is placed on a target place modified with chemical functionality; allowing some proteins attach to the modified surface. Then; a matrix molecule is applied to the surface to crystallize with the target peptides. Thus; the sample is analyzed with time-of-flight mass spectrometry where a laser ionizes the peptides from the samples. The resulting ions are accelerated through an electric field and the detector measures ions as they get to the end of the tube. The *m*/*z* ratio of each ion can be determined from the length of the tube; the kinetic energy given by the electric field and the time travelling the length of the tube. Some examples of modified surfaces used in SELDI are: CM10 (weak-positive ion exchange); H50 (hydrophobic surface; like C6-C12 reverse phase chromatography); IMAC30 (metal-binding surface); and Q10 (strong anion exchanger) [124]. Realistically speaking; SELDI is an older method that: only provides *m*/*z* reports; has data difficult to analyze and interpret; and lastly; does not produce identities. Consequently; it has been replaced with MALDI in most applications.

## 3. Proof of Concept for the Medical Field: The Successful Implementation of Mass Spectrometry in Evaluating Cancer

This comprehensive section is a roadmap of MS applications in medicine; focusing on diagnosis and prognosis in cancer; immunotherapy; investigation of proteins in laboratory; and clinical trials.

### 3.1. Introduction to Molecular Diagnosis for Cancer

Early diagnosis ensures possibilities of eliminating malignant tissues before metastasis to other organ systems. In 2000; National Cancer Institute established the Early Detection Research Network that relies on the development of biomarkers and technologies for the early diagnosis of cancer [125]. The progress of biomarkers panels for early cancer diagnosis is possible by improving the sensitivity and specificity of assays for tumors are in early development [126]. Detecting cancer at an early stage represents a challenge for researchers and medical personnel [127]. To be effective; screening tests must fulfill basic requirements such as a high degree of accuracy; detection at stage where the disease is curable; separation between aggressive lesions; requiring treatment; from harmless lesions; lastly; inexpensive; reproducible and correct calibrated to be of use [128,129].

Molecular diagnostic development represents the key in individualized medicine. Current diagnosis and prognosis classification based on anatomical; clinical methods do not highlight the heterogeneity of complex diseases; and cannot predict the therapy response or clinical outcome [130,131]. Based on these circumstances; the development of new molecular biomarkers that enhance diagnosis; evaluate the treatment response and identify disease progression [132,133,134] is in dire need. These molecular biomarkers could be altered genes; RNAs; proteins or metabolites which are specific for a pathological stage and offer information about molecular profiles of individual patients [135,136].

Nowadays; many studies strive to diagnose and treat different type of cancers at an early developmental stage. An important technique used in cancer diagnostics is proteomics that facilitates protein biomarker identification from blood samples (serum; plasma) [135,137,138]. Since biochemical blood characteristics are very variable and complex; it is very difficult to identify tumor-specific proteins like tumor antigens. Against these tumor antigens; antibodies are produced in an effective and specific biological amplification. It has been shown that this amplification begins at an early stage in carcinogenesis when the tumor is not clinically detectable [139,140]. The assays using antibodies for detection of tumor antigens utilize small amounts of tumor proteins that are easier to differentiate from the protein itself. Thus; tumor proteins become ideal for the detection of cancer at an early stage [141]. In addition; proteins released from tumors do not have a long half-time in the blood because they are rapidly cleared from the circulation in comparison with the antibodies that are highly stable in the serum [140,142].

A diagnostic assay relies on the development of antibodies; which involve important steps such as target-specific antigen discovery and identification; sensitive in vitro assay to detect antigen-specific antibodies; in vivo detection of antibodies from accessible body fluids (blood; saliva; urine); and assay validation for clinical use [135,136]. Thus; there are some immunodiagnostic assays used for disease diagnosis ranging from ELISA or individual point-of-care assays to fully automated multiplex immunoassay analysis [143,144,145]. A central high-throughput technology is microarray that analyzes the relative expression of genes simultaneously. Through microarray, genomic biomarkers can be detected that are tightly linked to cancer development. This type of technology can be applied to study uncommon categories of cancer patients called CUP [146]. The modification of genomic and transcriptomic patterns is often translated within proteomic profiles.

### 3.2. Early Cancer Diagnosis by Mass Spectrometry

In recent decades; an intense effort was directed toward the development of new therapies to enhance the survival rate of cancer patients. Unfortunately; the majority of patients with cancer are diagnosed in the late stages of the disease with regional or distant dissemination of cancer cells. However; when clinicians try to diagnose the cancer in the early stage when the tumor is limited; the survival rate can exceed 85% [147]. Mass spectrometry is being used as a diagnostic tool for different types of cancers such as breast; lung; pancreas; ovarian and prostate. In the following section details several current biomedical research results that are based on mass spectrometry. 

First up; Surface-enhanced Laser Desorption/Ionization (SELDI), a technique extending the base technology of mass spectrometry using a time-of-flight (TOF) analyzer to calculate the mass-to-charge ratio. SELDI-TOF-MS has great clinical diagnostic potential as a technology for detection or disease screening; as demonstrated for prostatic adenocarcinoma [148,149]. However; SELDI-TOF-MS should be used in combination with other diagnostic tools for verification or validation because it only underlies the differences in tumor and normal serum profiles by *m*/*z* reports and is limited in that it does not produce identities.

To determine the metabolic profiles for diagnostic purposes; ultra-performance liquid chromatography coupled with a quadrupole time-of-flight (UPLC-QTOF) mass spectrometer could be used. Matos Do Canto et al. utilized UPLC-QTOF to compare metabolic profiles in the ductal fluid from breast cancer patients with non-affected contralateral controls from the same subjects. In this way; they were able to differentiate the metabolites from tumor microenvironment where breast cancer forms. Moreover; they identified different metabolites with significant changes in the cellular microenvironment offering information about tumor evolution. Thus; this MS-based technique for metabolomic profiling aims is to distinguish early stage from advanced metastatic cancer and to develop a model for the early detection of breast cancer [150]. For oral squamous cell carcinoma; Wang Q et al. demonstrated that a comprehensive saliva metabolome analysis by an integrated MS-based method including reversed phase liquid chromatography (RPLC) and hydrophilic interaction chromatography (HILIC) can be used to identify biomarkers for early diagnosis. The authors identified fourteen potential biomarkers for early diagnosis of oral cancer where eight biomarkers were up-regulated and six down-regulated when compared to the healthy controls. More specifically; five biomarkers (propionylcholine; acetylphenylalanine; sphinganine; phytosphingosine; and S-carboxymethyl-L-cysteine) were proven to distinguish patients with oral cancer stages I-II from controls [151]. Furthermore; another metabolome study done by Budhu et al. revealed the presence of a specific metabolomic signature of tumors is dependent on the tissue origin. Each tissue and cancer type has a unique metabolites profile which was quantified and distinguished using gas chromatography combined with mass spectrometry (GC-MS) [152].

Moreover; Chen Y et al.; using the same technique GC-MS in their study; were able to differentiate the metabolite profiles of urinary samples from patients with gastric cancer. They distinguished a group of cancer metabolites; which exposed a diagnostic value for gastric cancer as important biomarkers for gastric cancer screening [153]. Lastly; Martinez-Garcia et al. developed an experimental design based on LC-PRM to verify endometrial cancer protein biomarkers in uterine aspirate samples. Firstly; they demonstrated that uterine aspirate samples are explicitly important for endometrial cancer protein biomarker screening. Secondly; they emphasize the benefits of high resolution mass spectrometry to determine and verify many potential biomarkers; thereby completing the information gap between discovery and validation studies [154].

### 3.3. Cancer Prognostic Evaluation Using Mass Spectrometry

Despite an increased worldwide life expectancy; the percentage of deaths caused by so-called non-communicable diseases; respectively cancer; heart disease and stroke; is increasing. Based on these statistics; the collective of researchers are sparking major efforts in discovery of novel and efficient biomarkers which could serve as better prognostic predictors of cancer. The rapid evolution and promising results from human proteome [155] and metabolomics studies [156] provide us with the expectation of improving prognostic instruments for these particular types of diseases.

Using mass spectrometry as a method of analysis provides high and consistent percentages of prognostic biomarker values achieved for the following types of cancers: breast; lung; colon; prostate; liver; pancreas; melanoma; thyroid; gastric and oral. In conducting these types of studies; various types of biological samples were analyzed; among which: plasma; serum; tissue; cultured cell lines; urine; saliva or seminal plasma [157]. A brief presentation of the applied MS-methodologies in the most relevant studies are presented in Table 3; emphasizing the cancer type; sample type; as well as whether the targeted proteins were validated.

The applicability of mass spectrometry in gynecological oncology has been a real success; where the outcome of prognosis is now over 90% in the evaluation of some patients diagnosed with ovarian cancer versus healthy subjects [155]. It was validated that calcyphosin identified by MALDI-QtoF MS represents a prognostic factor in the evaluation of patients diagnosed with endometrial cancer. Using MALDI-TOF MS for protein expression; after processing of serum collected from breast cancer patients versus healthy subjects; a five-protein panel was developed for prognosis; with a sensitivity and specificity over 85%. MALDI-IMS allows analysis; in short period of time; without pre-treatment/processing of the biological sample and for a large variety of solid tumors. The desire is to introduce this technique in clinical practice for prognostic purposes due to the possibility to receive immediate and reliable results. All of this being said; the implementation of this analysis requires standardization of protocols at large-scale and development of new tools of bioinformatics. Lastly; the application of MALDI-IMS in prognosis during in-surgery tissue characterization has a considerable impact on clinical research.

Depending on the different type of cancer; there is only a quantifiable patient response to immunotherapy in about 10–40% of the cases due to the fact that there are no accurate predictive biomarkers in routine use. The identification and development of predictive immune biomarkers can bring clear advantages to patients such as enhancement of the response rates in immunotherapy; decreased cost and toxicity caused by the chemotherapeutic agents [176]; improved diagnosis; evaluation of treatment efficacy; and determination of a personalized therapy with limited invasiveness [132]. This is imperative because many proteins or peptides are degraded by proteases before reaching in the bloodstream from leaky vessels [177]. Pathological changes can be identified by low molecular weight that provides some opportunities for clinical diagnosis or prognosis; and monitoring response to therapy [178].

### 3.4. Immunotherapy Assessment for Mass Spectrometry

Immunotherapy used for cancer treatment generally involves a wide range of concepts designed to activate the immune system or restore the anticancer immune response. Thus; recognition and elimination of cancer cells and the tumor microenvironment through an endogenous defense mechanism can be used to determine the stabilization and remission of the disease. In this way; cancer immunotherapy can enhance treatment efficiency through the combination of external treatments such as chemo-; radio-therapy or surgery with the internal defense [179]. Several strategies have been proposed for cancer immunotherapy trying to connect immune response stimulation against cancer cells with shifting immune balance from editing to elimination [180]. Based on patients’ immune response engagement; immunotherapy can be classified as active and/or passive. Thus; active immunotherapy stimulates the host immune system to produce an endogenous defense response for cancer cells: for example; utilizing cancer vaccines; adoptive cell transfer; adjuvants or checkpoint inhibitors. For passive immunotherapy; the host immune response is initiated by external compounds such as monoclonal antibodies and cytokines [179]. Recently; immunotherapy approaches have tried to boost the response of T cells and improve immune system activation of cancer cell destruction. This approach is represented by the immune checkpoint blockage which started with development of antibodies used against cytotoxic T lymphocyte-associated antigen 4; such as ipilimumab and tremelimumab. This development in immunotherapy continued with antibodies targeting the programmed death-1 (PD-1) T cell co-receptor; such as nivolumab and pembrolizumab; and its ligand B7-H1/PD-L1; such as durvalumab; atezolizumab; avelumab [181]. In conclusion; the aim of immunotherapeutic research is to utilize monoclonal antibodies for immune system stimulation to kill the remaining cancer cells after surgery and chemotherapy [182].

For different oncologic diseases there are some therapeutic antibodies; such as antibody-drug conjugates; that present a higher binding specificity and good pharmacokinetic for drug delivery targeting the cancer cells. Thus; the investigation into the primary structure of the antibodies is known as peptide mapping. This concept for mapping represents an important tool in understanding the primary structure of any therapeutic protein; amino acid composition; sequence variants; glycosylation heterogeneity and post translational modifications [183].

### 3.5. Current Protein Investigations Using Mass Spectrometry

Cell-surface proteins play a fundamental role in signal transduction; ion transport; cell adhesion and cancer pathogenesis. Their profile could offer a better understanding about cell-surface proteome; more explicitly how regulation occurs and their active response to different intracellular or extracellular signals. Consequently; cell-surface proteins could be exploited as targets for diagnosis; prognosis and therapy. A major drawback is the lack of characterization and quantification of tumor cell proteins that are critical for marker identification during clinical diagnosis and prognosis; or of drug targets for therapy. Regarding this aspect; the mass spectrometry technology has been developed as a specific platform that integrates cell-surface protein capture in vitro and intact protein fractionation for tumor cell protein analysis [184]. This makes mass spectrometry an indispensable tool for the cellular proteome analysis [185]; protein and peptide quantification [186]. To improve results for the target of interest; tandem mass spectrometry (MS/MS) has become a current demand for protein identification and quantification. To further improve results; it is also necessary to consider other important characteristics such as resolving power; mass accuracy and linear dynamic range. Moreover; fragment recognition through electron capture dissociation; electron transfer dissociation or collision-induced dissociation can identify different types of post-translational modifications [187].

Clinical classification of these protein molecules is measured as tumor markers; risk or prognostic markers; acute or chronic disease markers or hormone markers. Immunoassays are a common method used to detect protein analytes. In most cases; this method does not distinguish the different proteo-forms from the protein variants present in clinical sample [188]. Conversely; different proteo-forms can be detected with mass spectrometry after targeted protein enrichment. This enrichment improves the sensitivity and enhances quantification. Antibodies are specifically exploited for enrichment because of their ability to selectively target certain proteins even in protein rich matrices like plasma. Thus; utilizing antibody-coated beads for protein target enrichment; described in mass spectrometric immunoassay; facilitates the detection of intact proteins by MALDI-MS. Another variation could be to modify the MALDI target using antibodies; such is the case for Stable Isotope Standards and Capture by Antibody Peptide Antibodies (SISCAPRA) and immuno-MALDI [188]. For such techniques; antibody enrichment is done at the peptide level after digestion which greatly improves detection by MALDI-MS.

A recent article describes an immunoaffinity-based methodology of removing interfering high-abundant blood-derived proteins from human plasma and tissue samples [189]. Prieto et al. make the case that the use of mass spectrometry-based proteomics for the discovery of clinically relevant cancer biomarkers have proven challenging for one primary reason: the enormous dynamic range and high abundance of blood-derived proteins. Firstly; from a practical standpoint; they argue that the biomarker discovery phase should have simultaneous analysis of matched tissue and blood samples from one patient facilitating an improved identification of more authentic tumor proteins. Secondly; immunodepletion of the clinical tissue or fluid samples provides a reproducible solution by the removal of these highly abundant blood-derived proteins. The immunodepletion was achieved using the Agilent MARS Human 14 immunoaffinity cartridges; which are designed to chromatographically remove fourteen interfering high-abundant proteins from human plasma samples; these proteins include: albumin; IgG; antitrypsin; IgA; transferrin; haptoglobin; fibrinogen; alpha2-macroglobulin; alpha1-acid glycoprotein; IgM; apolipoprotein AI; apolipoprotein AII; complement C3; and transthyretin. The removal of these proteins effectively expands the range of the subsequent LC/MS and electrophoretic analysis of the samples. To further reduce the peptide complexity of the clinical sample and to maximize the coverage of each sample proteome; the authors demonstrated off-line Strong Cation Exchange(SCX)-fractionation could be used in shotgun proteomics [189].

To conclude this section; mass spectrometry encompasses all of the analytic laboratory demands for a fully integrated analysis from proteome profiling; biomarker discovery; protein degradation study; protein-protein interactions; protein-ligand interactions; and lastly; biological regulation through post-translational modifications [190].

### 3.6. Clinical Trials

Through the literature study conducted on website www.clinicaltrials.gov; it was determined that mass spectrometry is used in various ways for clinical trials; starting with the development of the drug and continuing with clinical trials in diagnosed patients vs. healthy volunteers (where is applicable).

Based on information obtained from approved clinical trial protocols; mass spectrometry; due to the accurate mass measurements based on ionic forms of samples which are separated according to *m*/*z* ratio of components; has the following qualifications for use. Firstly, and simply; biological sample analysis of tissues meaning sampling and comparing of cancer tissue to normal adjacent tissue. Secondly; biological sample analysis to determine substances/compounds of interest; which entails bioavailability/bioequivalence studies of analytes; evaluation of the relationship between oxidative DNA damage and use of antioxidants or mineral supplements; or determination of percentage of volatile organic compounds in the exhaled breath through advanced techniques like MALDI-TOF analysis (microscopic laser-directed protein mass spectrometric analysis) or rapid evaporative ionization.

It has been widely used in the evaluation of various types of cancer diagnoses; in different stages (Early Stage; Stage I and Stage II) such as: brain; breast; colon; endometrial; head and neck; lung; melanoma; ovarian; etc.

At present; even if the observational clinical trial is in the recruitment stage; the appropriate activities are being implemented to determine the advantages and disadvantages for the treated patients following therapies concordant with GPS Cancer test; offered by NantHealth. This revolutionary molecular test is integrating the most relevant quantitative measurements; respectively, the targeted proteomics detected by whole genome (DNA); mass spectrometry and whole transcriptome (RNA) sequencing; of both normal and cancer tissue. The results will greatly support oncology specialists in providing treatment; as a continuous promotion of personalized medicine and; last but not least; in decreasing costs associated with the care of cancer patients. The inclusion criteria for the study population; as per protocol requirements; contains full eligibility enrolment for metastatic pancreatic cancer; breast cancer patients; advanced stage lung cancer; metastatic colon cancer or metastatic non-resectable malignant melanoma at any time in their treatment history.

The types of MS analysis correlated with clinical trials are grouped into the following categories: dosage and purity analysis of the interest active substance; quantitative and qualitative analysis of the compound where evaluation occurs during the clinical study; routine analysis (according to the clinical trial protocol) for patients enrollment; or proper response evaluation corresponding to the approved clinical trial protocol (Figure 4).

However; researchers conducting the observational clinical trials still lament that patient recruitment remains a great challenge due to low percentage of participation (only 3–5% of diagnosed cancer patients participate in a clinical trial); 40% of studies failed to achieve minimum patient enrollment; fear of patients to reduce the quality of life; concerns involved receiving placebo; possible side effects; no-active encouragements from physicians; budgetary constraints of trial participation; the difficulty of eligibility according to inclusion/exclusion criteria; or lack of public education on emphasizing the importance of clinical trials.

## 4. Conclusions

Mass spectrometry has the potential to become a multi-platform analysis for all small molecules from biological samples; especially when coupled with other techniques like liquid chromatography to enhance sensitivity; selectivity; speed; molecular specificity and high-throughput chemical information. It is a tool for the identification and quantification of expression; modifications and interactions of proteins; powerful in offering understanding in cellular signaling pathways; detection of drug metabolites; diagnostics and drug discovery.

Using mass spectrometry; analyses which rely on the identification of cellular proteins and pathways causing DNA damage under the influence of chemotherapeutic agents; can be performed. Through this method; the protein; drug and metabolites from human serum or plasma can be detected and quantified. The advantages of this method include its enhanced sensitivity and selectivity while; concurrently; decreasing run time; elution time and consumption of solvent or sample. MS provides a more complete profile of regular and reactive metabolites; providing a great lead in drug metabolism monitoring and treatment streamline. All of this demonstrates that MS based techniques have been successfully implemented for omics studies; in particular for cancer research; meaning they could be extended to encompass broader medical studies as we adopt the concept of personalized medicine.

## Figures and Tables

**Figure 1 ijms-20-02576-f001:**
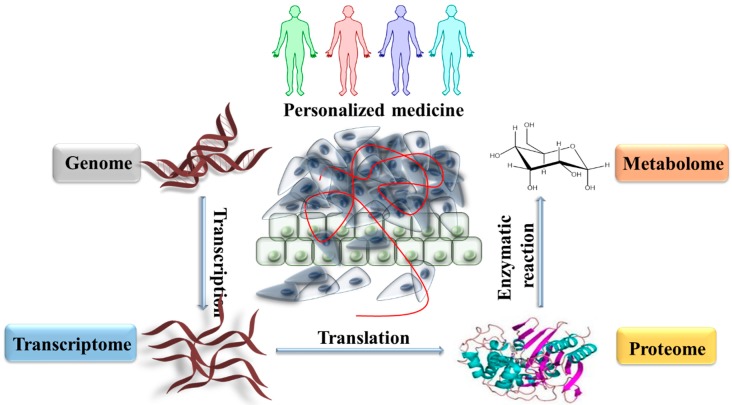
The complexity of cellular function involving associated omics technologies such as genomics; transcriptomics; proteomics and metabolomics. All the omics domains interact and influence each other. The interactions between the genome; transcriptome; proteome and metabolome within a tumor microenvironment are the building blocks that sustain cancer development and progression. By having a deep understanding over individual pathological changes that happen both inside every element; but also, at the level of their interactions; ontological therapeutic approaches could be developed in the future direction of personalized medicine. This approach could offer a far better match between cancer molecular type and therapeutic strategy.

**Figure 2 ijms-20-02576-f002:**
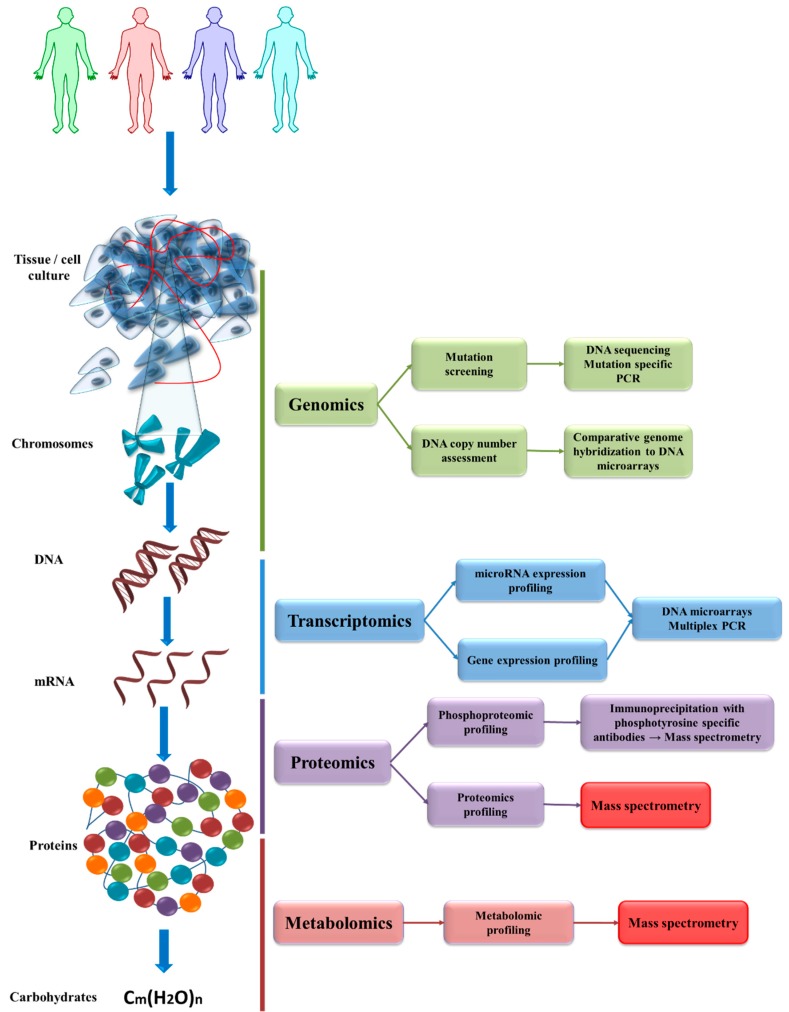
Diagram depicting the relationship between the major omics technologies. Both; the tissue and the isolated cell culture; are generally an in vitro replica of what happens inside a living organism. These biological samples contain chromosomes; which is where the DNA is extracted. The information contained at the DNA level is analyzed by genomic studies. The genomics look at the mutations found in the DNA or at the DNA copy number. The mutations are screened through DNA sequencing or mutation specific PCR. The increase or decrease in the copy number of different DNA regions is determined through comparative genome hybridization to DNA microarrays. The DNA is expressed by being transcribed into various RNA species. The RNA species constitutes the transcriptome. The transcriptome profile is determined by analyzing the microRNAs; or other non-coding/regulatory RNAs; and also coding mRNAs; meaning assessing the number of mRNAs of one kind. These are analyzed with the help of DNA microarrays or multiplex PCR. Some components of the transcriptome are translated into proteins. All the proteins found inside a living cell are analyzed with the help of proteomics. During the proteomics analysis; the profiling of a panel of proteins can be determined or their functional status can be analyzed with the help of phosphoproteomics. The mass spectrometry plays an essential role in proteomics profiling and in phosphoproteomics profiling. Different types of proteins with enzymatic function have complex interactions that build the metabolic pathways; resulting in different metabolites of endogenous or exogenous molecules. The analysis focusing on metabolites constitute the metabolomics profiling. Again; during the metabolomics profiling; the mass spectrometry plays an essential role. The aforementioned laboratory analysis can be transferred to a specialized; trained personnel and transformed into a standard clinical practice as a part of personalized medicine. Each patient has a unique set of changes in the genome; transcriptome; proteome and metabolome. These changes could lead to a different response to therapy and implicit adapting steps within the therapeutic approach. C_m_(H_2_O)_n_: C—Carbon; H—Hydrogen; O—Oxygen (where m may be different from n).

**Figure 3 ijms-20-02576-f003:**
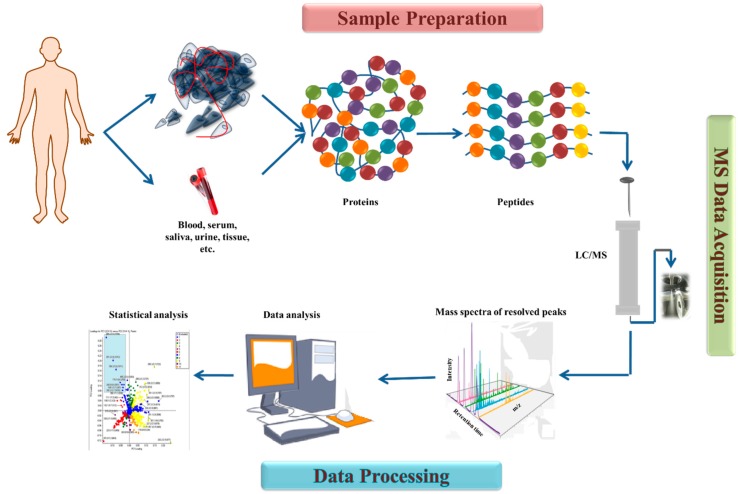
Mass spectrometry-based proteomics general work flow - from sample preparation through data analysis. Proteins are isolated from different types of biological samples taken from a subject/patient. The proteins are denatured and cleaved into fragments that have peptide structures; which are injected into the LC/MS/MS system; separated through the column; and the raw data are analyzed by the equipment software. The analyzed data are finally processed statistically and interpreted to correlate with the diagnostic; to lead to a personalized therapeutic approach.

**Figure 4 ijms-20-02576-f004:**
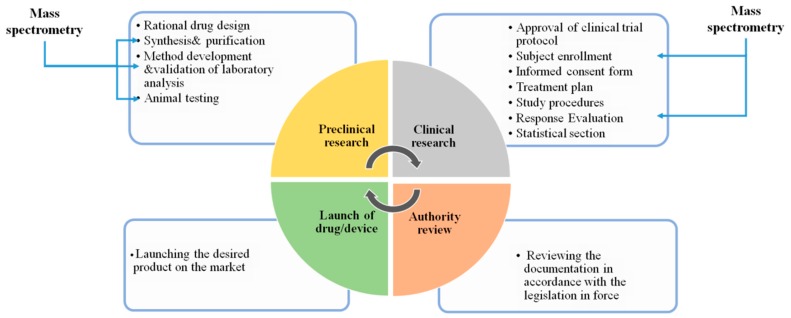
Flow Diagram emphasizing the importance of using mass spectrometry in the main stages of drug/medical device development.

**Table 1 ijms-20-02576-t001:** Omics applications in personalized medicine based on sample source; techniques and analysis available.

	Genomics	Transcriptomics	Proteomics	Metabolomics
**Definition**	Is relying on the studies of the structure; function and expression of the entire gene in an organism (coding and noncoding genes - miRNAs; siRNAs; piRNAs; snoRNAs; snRNAs; exRNAs; scaRNAs; lncRNAs; circRNAs).	Is relying on gene expression profiling; the evaluation of the mRNAs profiling that is present in a specific biological sample.	Focuses on proteins study and discovery; to understand the amount (quality and quantity) and functioning of proteins in biological systems.	Represent the measurement of all the metabolites in a biological sample-system.
**Sample Sources**	Genomic DNA and RNAs from all types of tissues.	mRNAs from all types of tissues.	All types of tissues and bio-fluids; most common used fluid is plasma.	Bio-fluids such as urine or plasma; tissue extract; in vitro cultures and supernatants.
**Techniques**	The methods used is sequencing of DNA segments that contain methylated fragments after DNA modification with sodium bisulfate or by genotyping using ‘genome-wide’ oligonucleotide arrays.	The most commonly used technique is Microarray that associates differences in mRNAs profiling from different groups of individuals to phenotype differences between the groups. This technique provides information about gene expression. Also; another important tool used is RNASeq that is used to study gene expression and identify new RNA species.	Identification of peptides/proteins can be determined using MS/MS based strategies. MS rely on three approaches: selection of protein spots from gels through 2-dimensional electrophoresis; gather abundance proteins and separate the less abundance by combining chromatographic approach; adsorb proteins using matrixes of immobilized chemicals based on charge; hydrophobicity; affinity; binding to specific ions following by desorption and MS/MS analysis.	MS –based techniques used to identify or determine the metabolites present are high-performance liquid chromatography; ultra-performance liquid chromatography; or gas chromatography.
**Analysis**	Bioinformatics methods (such as Annovar; Circos; DNAnexus; Galaxy; GenomeQuest; Ingenuity Variant Analysis; VAAST) are used to detect disease—association of gene; and genome analysis involved hierarchical clustering.	clustering is used to identify the gene sets and the data analysis used for gene interpretation can integrate microarray data with prior knowledge on the implication of genes in biological processes are needed (Gene spring; Feature extraction; R; Oncomine; Ingenuity Pathway Analysis, Hierarchical, DAVID Bioinformatics Resources; Panther).	Protein identification and analysis are performed by a variety of bioinformatic tools (such as Mascot; Progenesis; MaxQuant; Proteios; PEAKS CMD; PEAKS Studio; OpenMS; Predict Protein; Rosetta); which are available to researchers. Measurement (random) and systematic (bias) errors are necessary components of proteomic analysis.	To generate and interpret the metabolic profile of the sample; data generated are combined with multivariate data analysis such as partial least square; clustering; discriminant analyses (examples of metabolomic software; BioCyc -Omics Viewer; iPath; KaPPA-View; KEGG; MapMan; MetPa; Metscape; MGV; Paintomics; Pathos; Pathvisio; ProMetra).

Omics technologies might play a significant role in the generation of the etiology of disease and gene-environmental interactions [52,53,54]. Thus; omics can contribute to identify the exogenous risk factors that cause of disease and give some information about the interaction between macro environment and human health [53,54]. This step can be possible if we have an adequate validated technology; appropriate study designs and sample size; quality control and statistical methods for data analysis [55,56].

**Table 2 ijms-20-02576-t002:** List of Mass spectrometry analyses based on method; applications and advantages/disadvantages.

Mass Analyzer	Ionization Method	Applications (Examples)	Introducing the Sample	Technical Features of the System (Major Advantages/ Disadvantages)
TOF	MALDI EI; APCI; APPI	Protein identification using database library Peptide mapping Nucleotides	Solid matrix	No limit for mass determination and fast data acquisition Resolution better than; but sensitivity (quantitation) not as good as quadrupoles Exact masses with internal calibration
	SELDI	Protein mixtures analysis	Solid matrix	Is using chip surfaces Specific for low molecular weight of proteins (<20 kDaltons)
	MALDI	Proteins; peptides; lipids; small molecules from tissue (MALDI imaging)	Solid matrix	Detect a large amount of interest compound in a single run keeping intact the sample Major contributions in diagnostics; prognostics; drug delivery
Hybrid Quadrupole -TOF	ESI, APCI	Non-covalent interactions	LC or syringe	Exact masses with internal calibration Most sensitive full scan
Quadrupole	ESI; EI; APCI; MALDI	Scanning of parent-ion Study of ion-molecule reactions	LC or syringe	Nominal mass range: 0–4000 *m*/*z* Scan speed slow
Ion trap	ESI; APCI; MALDI	To acquire ions for subsequent analysis	LC or syringe	Lower costs and high accuracy in *m*/*z* determination Full scan medium
FTMS	ESI; APCI; EI	Label-free protein quantification	LC or syringe	Exact masses without internal calibration

There is no doubt that the recent development of mass spectrometry together with the superior specificity of the methods; is useful for different types of fields including chemistry; physics; medicine; or biology as an analytical robust tool; being the future choice in many clinical and research applications. Manufacturers are actively interested to develop equipments that are user-friendly; with high resistance for many analytical runs in clinical laboratory. [APCI—Atmospheric-pressure chemical ionization; APPI—Atmospheric-pressure Photoionization].

**Table 3 ijms-20-02576-t003:** List of the mass spectrometry methods from the most relevant studies investigating prognostic biomarkers based on different cancer types and the respective sample type obtained; including whether the targeted proteins were statistically validated.

Type of Cancer	Methodology	Type of Biological Samples	Validated Targeted Proteins	References
Breast	iTRAQ shotgun proteomics/SID-SRM and LF-SRM	tissue	Yes	[158]
	Transcriptomics/LRP-SRM	plasma	No	[159]
Colon	LF shotgun peptidomics/SID-SRM	urine	Yes	[160]
	SID-SRM	tissue	Yes	[161]
	Hyper-tex-SRM	tissue	Yes	[162]
Gastric	SID-SRM	tissue	Yes	[163]
Liver	LF shotgun peptidomics/SID-SRM	plasma	Yes	[164,165]
	Gel-based Proteomics/SID-SRM	serum	Yes	[166]
Lung	LF shotgun proteomics/SRM	Tissue	Yes	[167]
	SID-SRM	Plasma	Yes	[168]
Melanoma	SID-SRM	tissue	Yes	[169]
Oral	LRP-SRM	saliva	Yes	[170]
Pancreas	LF shotgun proteomics/LRP-SRM	tissue	Yes	[171]
Prostate	LF shotgun glycoproteomics/SID-SRM	Serum; tissue	Yes	[172]
	2D-DIGE-MS/SID-SRM	urine	Yes	[173]
	SID-SRM	Seminal liquid; blood plasma	Yes	[174]
Thyroid	LF- SRM; SID-SRM	tissue	Yes	[175]

Continuous research in this direction will ensure the improvement of personalized treatment attributed to patients diagnosed with cancer. [SRM = selected reaction monitoring; SID-SRM = Stable isotope dilution-selected reaction monitoring; LRP-SRM = Labeled reference peptide-selected reaction monitoring; LF-SRM = Label free- selected reaction monitoring; 2D-DIGE-MS = two-dimensional difference in gel electrophoresis-mass spectometry].

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
