# Peer review of "The Relevance of Mass Spectrometry Analysis for Personalized Medicine through Its Successful Application in Cancer “Omics”"

_ijms, 2019, doi:10.3390/ijms20102576_

Round 1
Reviewer 1 Report
The review 'the relevance of mass spectrometry analysis for personalized medicine throught its successful application in cancer 'omics' is a summary of mass spectrmetric applications in the medical and clinical research area.
The revised form of review improved the previous version of manuscript. Indeed the author have provided more technical and scientific details to made the new version of the manuscript more satisfactory
Author Response
We would like to thank the reviewer for their time and effort spent reading the manuscript, as well as offering excellent feedback to improve it. Thank you again for considering our work and we hope that this new version is superior.
Reviewer 2 Report
This article describes a review of several methods for the biomarkers analysis in cancer using MSMS. Some minor changes and issues could still improve the paper:
- Page 2, figure 1: figure legend is too long. It is well explained by itself, but maybe all this science could be explained in the text.
- Page 7, table 1: NMR is mentioned only once in the manuscript, in this table. I would recommend explain the improvements and advantages of MS over NMR for cancer (for example, in section 2.1)
- Page 9, line 292-295: I disagree with “it is not mandatory to pre-treat or process the biological sample”. Maybe is it true for NMR, not for MS. Please, make a reference with a paper if still agree with this.
Revise the English of line 292 “It is a major advantage is to have several”
- Page 10, table 2: ESI is written with TOF, but it does not use Solid matrix. I would add LC and Solid matrix.
- Page 12, line 364: sometimes liquid extraction with organic solvents to protein precipitation is enough in Metabolomics. Try to comment these aspects through the text.
- Page 18, table 3: Is not possible to add the references also to the table in order to not search the articles in the text? Please, try to configure the figures and tables explaining them by themselves (but not too long figure captions as Figure 1 and 2).
- Page 19: Please, revise these recent articles and consider to be added to Section 3.5: https://www.ncbi.nlm.nih.gov/pmc/articles/PMC4201940/
https://www.ncbi.nlm.nih.gov/pubmed/31077580
Author Response
May 21, 2019
Article Revisions:
Journal: International Journal of Molecular Medicine
Title: The relevance of mass spectrometry analysis for personalized medicine through its successful application in cancer “omics”
Author List: Cristina Alexandra Ciocan-Cȃrtițǎ, Ancuța Jurj, Mihail Buse, Lajos Raduly, Diana Gulei, Cornelia Braicu, Roxana Cojocneanu, Lavinia Lorena Pruteanu, Cristina Adela Iuga, Ovidiu Coza and Ioana Berindan-Neagoe
We would like to thank all of the reviewers for their time and effort spent reading the manuscript, as well as offering excellent feedback and interesting literature suggestions to improve it. We took the feedback back critically, trying to address each comment as directly and comprehensively as possible. More specifically, we have written more paragraphs, including added more references respectively, relating to the literature suggestions. Thank you again for considering our work and we hope that this new version is superior.
The following bullet points are a list of the weaknesses of the manuscript indicated by the reviewers:
· Comment 1: Page 2, figure 1: figure legend is too long. It is well explained by itself, but maybe all this science could be explained in the text.
Response: Thank you for indicating that the Figure text was too long. To address this, the main body of the Figure 1 explanation was moved to page 2 line 48-65: “The genome contains the basic information that can be inherited, or it can be modified throughout a person’s lifetime. Therefore, it contains both coding and non-coding DNA regions that are transcribed into various types of RNA species. All RNA types constitute the transcriptome, this entails that the transcriptome also contains both coding and non-coding RNA species. The non-coding RNAs fulfill various roles in the cell, from ribosomal constituent or transportation to gene expression regulation. The coding RNAs are translated into proteins. All the proteins within a living cell are grouped under the name of proteome. The functional unit of a cell is its proteins, determined by its structure which does not remain constant. Proteins are unique in that their structure can be post-translationally modified, thereby, altering their function. This means that proteins have more than one state which through inactivity (indirectly) or activity (directly) determine their functional use within the cell. The most common post-translational modification (PTM) that occurs is phosphorylation, turning certain enzymes into an active or functional for cellular energy transfer purposes. Regardless of the post-translational modification, the role that non-functional proteins have is not been completely elucidated. The various proteins and their isoforms have highly complex interactions with each other. Through the enzymatic interaction of various types of proteins, metabolic pathways are built. These pathways determine how endogenous or exogenous molecules will be processed in the living cell. The molecules resulting after the enzymatic reactions between proteins are named metabolites.”
· Comment 2: Page 7, table 1: NMR is mentioned only once in the manuscript, in this table. I would recommend explain the improvements and advantages of MS over NMR for cancer (for example, in section 2.1)
Response: Thank you for the observing that nuclear magnetic resonance appears in the table and article without being explicitly discussed. Based on your suggestion and considering that NMR is a complex approach which demands a comprehensive description and space to be presented properly, we decided to remove it from article.
The following sentences were modified to reflect this change:
Page 7, under the “Metabolomics” column of Table 1 “MS –based techniques used to identify or determine the metabolites present are high-performance liquid chromatography, ultra-performance liquid chromatography, or gas chromatography.”
· Comment 3: Page 9, line 292-295: I disagree with “it is not mandatory to pre-treat or process the biological sample”. Maybe is it true for NMR, not for MS. Please, make a reference with a paper if still agree with this. Revise the English of line 292 “It is a major advantage is to have several”
Response: Thank you for pointing out this grammatical error and misunderstanding in protocol. After having verified with our proteomics department, we agree that MS does require pre-treatment of the biological samples. Subsequently, we changed the first advantage of having several variations to MS, to better express the aforementioned sentiment.
The following sentences were corrected:
Page 9 Lines 290-296 “Protocols differ depending on the biological matrix and type of experiment, which includes the aim of the investigation. There are major advantages to having several variations of mass spectrometry because: firstly, implementing multiple pre-treatment protocols of the biological samples improves our understanding and experimental efficacy by reducing the protein abundance of the analyte; secondly, the equipment permits a direct scanning and analysis of the sample; and lastly, different variations allow for cross-validation of the data.”
· Comment 4: Page 10, table 2: ESI is written with TOF, but it does not use Solid matrix. I would add LC and Solid matrix.
Response: Thank you for identifying this mistake in Table 2. Therefore, ESI was removed from the TOF row.
· Comment 5: Page 12, line 364: sometimes liquid extraction with organic solvents to protein precipitation is enough in Metabolomics. Try to comment these aspects through the text.
Response: Thank you for providing an additional extraction method relating to the sample preparation, specifically with reference to Metabolomics. We particularly addressed this concern in Section 2.2 with the following paragraph:
Page 12 Lines 372-388 “Additionally, the use of organic solvents for the extraction of proteins and peptides from serum is efficient and facilitates the identification and comparison using different mass spectrometry approaches. Chertov et al. developed such a method, under denaturing conditions using acetonitrile containing 0.1% trifluoroacetic acid, for the extraction of peptides and low molecular weight proteins from serum samples. Using an older approach, called Surface-enhanced laser desorption/ionization-time of flight mass spectrometry (SELDI-TOF MS), they were able to detect two significantly reduced markers in mice with B cell lymphoma tumors; Apolipoprotein A-II was identified as one of these markers. It should be noted that liquid extraction with the use of organic solvents is used more frequently in Metabolotics. More specifically, if the metabolites are moderately polar, non-polar or hydrophobic, organic solvents must be included. It offers a unique advantage when compared to water in that more diverse metabolites can be extracted. For example, when implementing a two phase solvent system with water-methonal-chloroform both polar, hydrophilic or non-polar metabolites can be extracted simultaneous. Furthermore, organic solvent-based extractions also offer the advantages of: easy solvent evaporation, absence of precipitate salts and increased stability of extracted metabolites. Last but certainly not least, the organic solvents used for extraction are compatible with GS-MS, LC-MS, HPLC and capillary electrophoresis.”
· Comment 6: Page 18, table 3: Is not possible to add the references also to the table in order to not search the articles in the text? Please, try to configure the figures and tables explaining them by themselves (but not too long figure captions as Figure 1 and 2).
Response: Thank you for highlighting the absence of references from Table 3, it appears that they were accidentally cut from the original table. Table 3 was modified by: changing the column head from “Number of Targeted Proteins” to “Validated Protein Targets”; and adding a “Reference” column with the respective citations. Furthermore, a more descriptive title was added to Table 3:
Page 18 Lines 668-670 ”List of the mass spectrometry methods from the most relevant studies investigating prognostic biomarkers based on different cancer types and the respective sample type obtained, including whether the targeted proteins were statistically validated.”
Furthermore, as indicated Lines 665-667 on page 18 were modified to be more specific to the reason we presented of the table: “A brief presentation of the applied MS-methodologies in the most relevant studies are presented in Table 3, emphasizing the cancer type, sample type, as well as whether the targeted proteins were validated.”
· Comment 7: Page 19: Please, revise these recent articles and consider to be added to Section 3.5:
https://www.ncbi.nlm.nih.gov/pmc/articles/PMC4201940/
https://www.ncbi.nlm.nih.gov/pubmed/31077580
Response: Thank you very much for the recent literature suggestion, both articles offered interesting critical analysis addressing the reproducibility of protein biomarkers from mass spectrometry approaches. After reading the articles, we hope that you are in agreement with using only one of the articles by Prieto et al. (2019) titled “Mass spectrometry in cancer biomarker research: a case for immunodepletion of abundant blood-derived proteins from clinical tissue specimens.” The second article by Liang et al. (2019) titled “Reproducibility of biomarker identifications from mass spectrometry proteomic data in cancer studies”, albeit an important topic, we believe extends beyond the scope of the manuscript. In essence, to address the problem of consistent or standardized data analysis and statistics of mass spectrometry proteomic data, a full discussion needs to be had and making quick reference to it in this manuscript does not give its importance justice.
The following sentences with reference to the Prieto et al. article were added to Section 3.5:
Page 20 Lines 758-775 “A recent article describes an immunoaffinity-based methodology of removing interfering high-abundant blood-derived proteins from human plasma and tissue samples. Prieto et al. make the case that the use of mass spectrometry-based proteomics for the discovery of clinically relevant cancer biomarkers have proven challenging for one primary reason: the enormous dynamic range and high abundance of blood-derived proteins. Firstly, from a practical standpoint, they argue that the biomarker discovery phase should have simultaneous analysis of matched tissue and blood samples from one patient facilitating an improved identification of more authentic tumor proteins. Secondly, immunodepletion of the clinical tissue or fluid samples provides a reproducible solution by the removal of these highly abundant blood-derived proteins. The immunodepletion was achieved using the Agilent MARS Human 14 immunoaffinity cartridges, which are designed to chromatographically remove fourteen interfering high-abundant proteins from human plasma samples; these proteins include: albumin, IgG, antitrypsin, IgA, transferrin, haptoglobin, fibrinogen, alpha2-macroglobulin, alpha1-acid glycoprotein, IgM, apolipoprotein AI, apolipoprotein AII, complement C3, and transthyretin. The removal of these proteins effectively expands the range of the subsequent LC/MS and electrophoretic analysis of the samples. To further reduce the peptide complexity of the clinical sample and to maximize the coverage of each sample proteome, the authors demonstrated off-line Strong Cation Exchange(SCX)-fractionation could be used in shotgun proteomics.”
Given all these additions and reflection, we decided to modify some of the sub-heading of Section 3:
Page 16 Line 570 “3.1. Introduction to Molecular Diagnosis for Cancer”.
Page 17 Line 610 “3.2. Early Cancer Diagnosis by Mass Spectrometry”.
Page 18 Line 654 “3.3. Cancer Prognostic Evaluation using Mass Spectrometry”.
Page 19 Line 699 “3.4. Immunotherapy Assessment for Mass Spectrometry”.
Page 20 Line 728 “3.5. Current Protein Investigations using Mass Spectrometry”.
Final comment: We thank you for and appreciate the suggestions given to improve the manuscript to the level of quality attributed to this journal.